# Retrieval-Augmented Parsing for Complex Graphs by Exploiting Structure and Uncertainty

**Zi Lin**
UC San Diego
lzi@ucsd.edu

**Quan Yuan**
Google Research
yquan@google.com

**Panupong Pasupat**
Google Research
ppasupat@google.com

**Jeremiah Liu**[†‡]
Google Research & Harvard University
jereliu@google.com

**Jingbo Shang**[†]
UC San Diego
jshang@ucsd.edu

## Abstract

Retrieval augmentation enhances generative language models by retrieving informative exemplars relevant for output prediction. However, in realistic graph parsing problems where the output space is large and complex, classic retrieval methods based on input-sentence similarity can fail to identify the most informative exemplars that target graph elements the model is most struggling about, leading to suboptimal retrieval and compromised prediction under limited retrieval budget. In this work, we improve retrieval-augmented parsing for complex graph problems by exploiting two unique sources of information (1) structural similarity and (2) model uncertainty. We propose *Structure-aware and Uncertainty-Guided Adaptive Retrieval* (**SUGAR**) that first quantify the model uncertainty in graph prediction and identify its most uncertain subgraphs, and then retrieve exemplars based on their structural similarity with the identified uncertain subgraphs. On a suite of real-world parsing benchmarks with non-trivial graph structure (SMCalflow and E-commerce), SUGAR exhibits a strong advantage over its classic counterparts that do not leverage structure or model uncertainty.

## 1 Introduction

Large language models (LLMs) have demonstrated remarkable capabilities as effective few-shot learners (Brown et al., 2020). Recently, a new learning paradigm called in-context learning has been developed. Under this paradigm, the model is given a *prompt* including test inputs and a few related *exemplars*, and can generate outputs directly without updating its parameters. A typical approach to obtaining these exemplars is to retrieve training examples similar to the test input (Pasupat et al., 2021; Gupta et al., 2022). However, for realistic parsing tasks with large output graphs and non-trivial structure (e.g., dialogue-oriented semantic parsing), the

input similarity alone may not be effective in identifying the most informative exemplars for aiding graph prediction. As an example mentioned in Qiu et al. (2022), for the test input "Schedule a meeting with my manager", it is more similar to example "Schedule a meeting with Alex" than "Who is my manager", while the latter one contains an important action for searching an org chart which is also required by the test input.

In this paper, we explore effective approaches to improve the generalization performance of retrieval-augmented LLMs for parsing complex graphs. Specifically, we consider exploiting two sources of information uniquely available to this problem: (1) structural similarity between output subgraphs, and (2) model uncertainty in graph component prediction. The motivations behind our approach are two empirical investigations on LLM graph parsing (presented in Section 3): (a) *Inadequacy of sequence-similarity retrieval* (Section 3.1). When output graphs exhibit non-trivial structure, the exemplars retrieved based on sequence similarity is less effective than those based on graph similarity, even when the similarity is computed with respect to the gold output graphs [1]. (b) *LLM uncertainty correlates with performance* (Section 3.2). We conduct a exploratory study of the quality of LLM uncertainty as an indicator of its generalization performance in graph prediction, and identified a monotonic between model uncertainty v.s. accuracy for node and edge prediction (Figure 2). This implies that model uncertainty can serve as an effective signal for identifying the subgraphs that the model is struggling to predict, thereby helping the retrieval algorithm to efficiency identify the most effective examplars for aiding model prediction, especially when the output graph is large.

Based on the above observations, we propose

---

† Co-senior authors. ‡ Work done at Google.

[1]This is in contrast to earlier work where sequence similarity is already sufficient for simple sentences with shallow output structures (e.g., TOPS) (Zemlyanskiy et al., 2022)

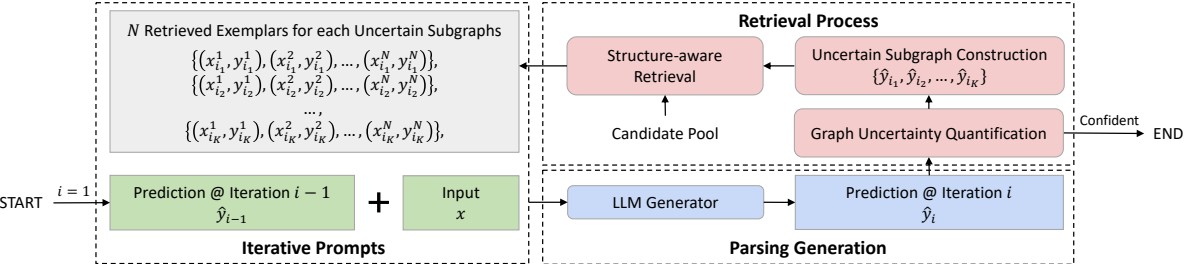

Figure 1: An overview of **S**tructure-aware and **U**ncertainty-**G**uided **A**daptive **R**etrieval (**SUGAR**). At each iteration $i$, the model prediction $\hat{y}_i$ will go through a retrieval process consisting of three steps (red squared ones): (1) **graph uncertainty quantification** that measures the model's uncertainty in predicting structured output $y_i$ at the sub-structure level (Section 4.1.1); (2) **uncertain subgraph construction** that gets a collection of uncertain subgraphs $\hat{y}_{i_k}$ based on uncertainty scores of $y_i$ (Section 4.1.2); (3) **structure-aware retrieval** that retrieves structurally similar exemplars targeting at $\hat{y}_{i_k}$ (Section 4.1.3). Note that at iteration 0, the exemplars are retrieved by input similarity since there is no model prediction yet.

*Structure-aware and Uncertainty-Guided Adaptive Retrieval* (**SUGAR**), a retrieval-augmented LLM inference framework for complex graph parsing that incorporates both *structural similarity* and *model uncertainty* into retrieval procedure (Section 4). Operating in an iterative manner, SUGAR first identifies uncertainty regions in the model's graph prediction from the previous iteration, and adaptively retrieves exemplars based on their graph similarity with the identified uncertainty subgraphs (Figure 1, Section 4). In this way, SUGAR is able to better target model weaknesses in structural prediction by retrieving the most informative exemplars, which appears to be especially valuable in the setting of large and complex output graphs given limited retrieval budget.

On a suite of real-world complex graph parsing benchmarks (i.e., SMCalFlow and Ecommerce), SUGAR exhibits distinct strength over its classic counterparts that do not leverage uncertainty or structure, bringing clear performance improvement over the base model across iterations even when other retrieval-augmentation methods become counterproductive (Section 5). Further in-depth analysis revealed the interesting role of exemplar quality on model uncertainty, the effectiveness of uncertainty as an early-stopping signal for retrieval iterations, and verifies the effectiveness of structural retrieval in improving model confidence (Section 6)[2].

## 2 Related Work

**Retrieval-Augmented Parsing.** Sequence-to-sequence (seq2seq) models have achieved state-of-the-art performance on many natural language processing tasks including complex parsing, e.g., dialogue-oriented semantic parsing and meaning representation parsing (Vinyals et al., 2015; Xu et al., 2020; Cui et al., 2022; Lin et al., 2022b,a). The general approach is to treat the output structure as a sequence and fine-tune a seq2seq model to learn the mapping between input sentences and output structures. To reduce the reliance on large-scaled annotated data, several work augment the input with retrieved exemplars from the training data, with different strategy to select exemplars.

For unsupervised retrieval, Pasupat et al. (2021) and Gupta et al. (2022) retrieve exemplars with similar input encodings from a pre-trained neural encoder. Zemlyanskiy et al. (2022) retrieves exemplars for which the input and output (from the preliminary prediction) has high TF-IDF similarity with the sample input. The above work mainly focused on fine-tuning settings. For supervised retrieval, Rubin et al. (2022) suggest to use language models themselves to label examples that can serve as good prompts, and train a prompt retriever from this signal. In this work, we focus on unsupervised retrievers that do not rely on additional training data beyond the candidate pool they retrieve from.

**Iterative Retrieval.** While LLMs can generate coherent outputs via single-time retrieval-based augmentation, they often fall short in addressing more complex tasks. To address this, there have been various attempts to retrieve exemplars more than one time. Most of the work focused on ad-

---

[2]Open-source code may be found at https://github.com/google/uncertainty-baselines.

| Metric | Similarity | MTOP | | | SMCalFlow | | | Ecommerce | | |
|---|---|---|---|---|---|---|---|---|---|---|
| | | StoO | EM | SMATCH | StoO | EM | SMATCH | StoO | EM | SMATCH |
| USE | Input | 68.64 | 44.60 | 87.08 | 48.38 | 24.20 | 68.59 | 45.79 | 5.10 | 66.53 |
| BM25 | Input | 57.08 | 41.40 | 84.34 | 51.33 | 24.20 | 72.78 | 47.21 | **8.40** | 69.64 |
| BM25 | Output | 87.47 | **50.90** | 94.09 | 71.93 | 48.10 | 89.03 | 51.09 | 8.00 | 69.74 |
| SMATCH | Output | **90.09** | 50.50 | **94.10** | **80.16** | **53.80** | **92.03** | **60.45** | 7.30 | **72.88** |

Table 1: Evaluation results for in-context learning with 10 exemplars retrieved based on different similarity functions. StoO means average SMATCH to oracle which measures the graph overlapping between gold output and retrieved outputs; EM means exact match rate.

dressing long-form outputs such as long-form question answering tasks (Fan et al., 2019; Stelmakh et al., 2022). The basic idea is to decomposing complex question into several easier sub-questions, and iteratievely retrieving relevant information from knowledge agents for each sub-quesitons (Press et al., 2022; Yao et al., 2022; Khot et al., 2022). Based on this line of work, FLARE (Jiang et al., 2023) further proposes to actively retrieving when the sub-answer contains low-confident tokens.

However, iterative retrieval for parsing complex structured outputs is less explored. The core challenge lies in the non-sequential nature of the output structure such as tree or graph, which means it cannot be simply decomposed sequentially. In this work, we aim at addressing complex parsing tasks, and target on progressively improving model's prediction by iteratively retrieving relevant examplars for model's uncertain sub-structures. As we will show in Section 3, this can not be achieved without incorporating structure and uncertainty.

## 3 Motivations

In this section, we present two ablation studies which serve as motivations for our methods.

### 3.1 Structural Similarity Matters

The first question is *what to retrieve*. Here we study how different similarity functions perform on different semantic parsing tasks, which is under in-context learning settings using GPT3.5 with 10 exemplars in the prompt.

Specifically, we test input sentence similarity using Universal Sentence Encoder (USE) (Cer et al., 2018) and BM25 (Schutze et al., 2008), and output similarity using BM25 and SMATCH (Cai and Knight, 2013). Note that SMATCH is the only metric that considers structural similarity beyond simple token overlapping in the output (more details in Appendix A). We choose three different

semantic parsing tasks with output structures from simple to complex, including (1) MTOP (Li et al., 2021): a user intent slot filling dataset which can be simplified as a sequence labeling task; (2) SMCalFlow (Andreas et al., 2020): a dataset of semantically detailed annotations of task-oriented natural dialogues, which can be taken as a tree parsing task; (3) Redwoods-Ecommerce (Ecommerce for short) (Oepen et al., 2002): a dataset of annotated meaning representation (outputs are directed acyclic graphs) based on English Resource Grammar (Flickinger et al., 2014), which is a DAG parsing task.

The evaluation results are shown in Table 1. We can observe gaps between standard retrieval (based on input similarity) versus oracle retrieval (based on output similarity), a finding that aligns with Qiu et al. (2022). Furthermore, as the output structure gets more complex (from MTOP to Ecommerce), it becomes more important to have exemplars that are similar in output structure, compared to just input similarity or sequence-level output similarity. This is because sequence-level similarity metrics only consider token overlapping, and ignore syntactic or semantic relationships in the output structure. As output becomes more complex, these similarity metrics are less likely to find structural similar exemplars. Therefore, it is important to have a structure-aware retriever for complex parsing tasks.

### 3.2 Model Uncertainty Matters

However, retrieving exemplars based structural similarity can have several challenges. First, in the initial settings we do not have access to gold output structures. This can be solved by getting some preliminary prediction using retrievals with similar inputs as proposed in Zemlyanskiy et al. (2022). Second, retrieving similar outputs based on the entire structures can be ineffective and may introduce unwanted noise. Given the emerging challenges, we are investigating if it is possible to measure struc-

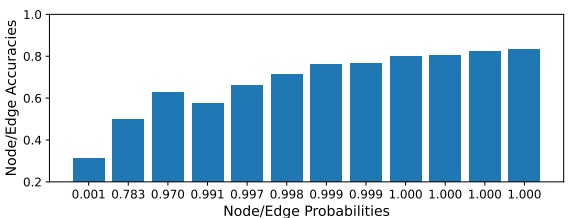

Figure 2: Correlations between model probabilities and performance for node/edge prediction

tural similarity only on necessary sub-structures. If so, *which part to retrieve*? Our hypothesis is that we can retrieve only when model is uncertain about some sub-structure predictions, which are very likely to be flawed.

To validate our hypothesis, our second ablation study analyzes LLM's behavior in predicting structure components in terms of model uncertainty. Specifically, we explore the correlation between model probability and performance for in-context learning model (more details can be found in Appendix B). As shown in Figure 2, high model probability generally corresponds to high performance and vice versa. Our study confirms that model uncertainty is effective for detecting prediction errors. This means that we can retrieve structurally similar exemplars targeting on these uncertain substructures, which can help to address the flawed parts in the prediction.

## 4 SUGAR: Structure-aware and Uncertainty-Guided Adaptive Retrieval

This section describes details of SUGAR for parsing complex structures. Typically, the output structure is a semantic graph that is rooted, directed, acyclic and labeled (Opitz, 2023).

**Problem Formulation.** We aim to solve a graph parsing problem that maps from a natural language utterance $x$ to a target graph representation $G = \langle \mathbf{N}, \mathbf{E} \rangle$, where $\mathbf{N}$ is the set of nodes and $\mathbf{E} \in \mathbf{N} \times \mathbf{N}$ is the set of edges. For seq2seq models, $G$ is represented as a linearized graph string $y$. Following Lin et al. (2023), we adopt PENMAN annotation (Kasper, 1989) to linearize all graph structures in this work, which is a serialization format for the directed, rooted graphs used to encode semantic dependencies (details for graph linearization can be found in Appendix C).

Figure 1 shows an overview of SUGAR and Algorithm 1 summarizes the detailed process. In Section 4, we illustrate the retrieval process based

---

**Algorithm 1** Retrieval-augmented Inference with SUGAR

**Precompute:**
  Examplar pool $P = \{(x_c, y_c)\}_{c=1}^{|P|}$
  Subgraph pool $P_g = \cup_c S_d(y_c)$    ▷ (Sec. 4.1.3)
**Input:**
  Test input $x$
**Output:**
  Final graph prediction $\hat{y}_i$
**Initialize:**
  $E_0 = \text{base\_retriever}(x)$    ▷ Initial retrieval.
  $\hat{y}_0 = \text{LLM}(x, E_0)$    ▷ Initial prediction.
# Iterative retrieval with early stopping.
**for** $i \in [1, ..., \texttt{max\_iter}]$ **do**
  # Graph uncertainty quantification (Sec. 4.1.1).
  $\{p_v\}_v = \text{graph\_component\_probability}(\hat{y}_{i-1})$
  # Confidence-based early stopping.
  **if** $p_v > \texttt{conf\_threshold} \ \forall v$ **then**
    **return** $\hat{y}_{i-1}$
  # Construct uncertainty subgraphs (Sec. 4.1.2).
  $\{\hat{y}_{i_k}\}_k = \text{get\_uncertain\_subgraph}(\{p_v\}_v)$
  # Structure-aware retrieval (Sec. 4.1.3).
  $E_i = \varnothing$
  **for** $\hat{y}_{i_k} \in \{\hat{y}_{i_k}\}_k$ **do**
    $E_{ik} = \text{structure\_aware\_retrieval}(\hat{y}_{i_k}, P_g))$
    $E_i.\text{add}(E_{ik})$
  # Retrieval-augmented prediction.
  $\hat{y}_i = \text{LLM}(x, E_i, \hat{y}_{i-1})$

---

on preliminary predictions $\hat{y}_i$ at step $i$ (the retrieval process in Figure 1), which includes three steps: (1) graph uncertainty quantification for $\hat{y}_i$ (Section 4.1.1); (2) uncertain subgraph construction, i.e., $\hat{y}_{i_k}$ (Section 4.1.2); (3) structure-aware retrieval for $\hat{y}_{i_k}$ (Section 4.1.3). The retrieval process is operated in an iterative manner, which enables a progressive improvement for model's prediction by iteratively retrieve exemplars based on model's predictive uncertainty from the previous turn (Section 4.2).

### 4.1 Retrieval Process

#### 4.1.1 Graph Uncertainty Quantification

Recent year witnessed the success of applying seq2seq models to graph parsing tasks, where the outputs are compositionally structured (e.g., a graph or a tree). However, these seq2seq approaches pose a technical challenge in properly quantifying the model uncertainty for graph prediction. This is because the autoregressive seq2seq probability is not well-suited for describing model uncertainty in predicting elements or substructures of the output graph, where the probabilistic graphical model (PGM) is a more suitable formalism. To address this issue, we leverage Graph Autoregressive Process (GAP) proposed by Lin et al. (2023) to allow the correspondence between seq2seq output probability to PGM probability, i.e., assigning

model probability for a node or edge on the graph.

Specifically, given an input sequence $x$ and output sequence $y = y_1 y_2 \cdots y_N$ that refers to a graph $G = \langle \mathbf{N}, \mathbf{E} \rangle$, GAP can map the token-level autoregressive distribution

$$p(y|x) = \prod_{i=1}^{N} p(y_i|y_{<i}, x)$$

to graphical model likelihood

$$p(G|x) = \prod_{v \in G} p(v| \operatorname{pa}(v), x)$$
$$= \prod_{n \in \mathbf{N}} p(n| \operatorname{pa}(n), x) * \prod_{e \in \mathbf{E}} p(e| \operatorname{pa}(e), x)$$

where $p(v| \operatorname{pa}(v), x)$ is the conditional probability for graph elements $v$ with respect to their parents $\operatorname{pa}(v)$ in $G$.

### 4.1.2 Uncertain Subgraph Construction

To leverage uncertainty in the model prediction $p(G|x)$ for efficient retrieval, we consider the concept of *uncertain subgraph* which is a subgraph that contains:

- **Uncertain element.** We consider a graph element $v \in G$ to be uncertain if its probability $p(v| \operatorname{pa}(v), x)$ is below a certain threshold $\epsilon$.

- **Relatively-confident neighbors.** Given a uncertain element $v$ and a subgraph $s_d(v)$ that surrounds $v$ and with maximum depth $d$. We define the *relatively-confident neighbors* of $v$ as the subset $c_d(v) \subset s_d(v)$ whose probability is above a certain threshold $\epsilon$.

As shown, by coupling the uncertain element $v$ with its relatively-confident graph neighbor $c_d(v)$, uncertain subgraphs $\hat{y}_v = \{v\} \cup c_d(v)$ provides the retrieval algorithm fine-grained and contextualized information about model uncertainty in the prediction of graph elements (see Figure 3 for an example). In practice, to limit the size of uncertain subgraphs and keep the cost of structural similarity computation within a feasible range, we set a parameter $d$ to control the maximum depth of uncertain subgraphs $\hat{y}_v$.

### 4.1.3 Structure-aware Retrieval

To identify informative graph exemplars that best address the model uncertainty in predicting graph elements, we leverage the uncertain graph introduced above and consider a retrieval policy using

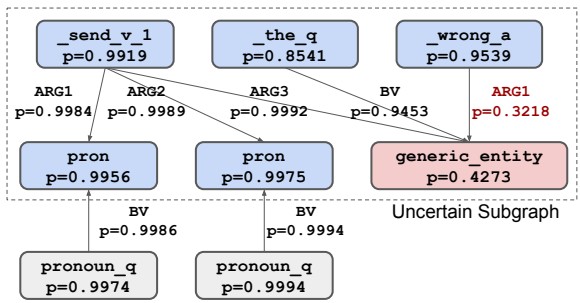

Figure 3: An example for uncertain subgraph ($\epsilon = 0.8, d = 2$) in prediction for input sentence "You send me the wrong camcorder" (Ecommerce). The dotted squared one is the uncertain subgraph, where the red squared node and red fonted edge refer to uncertain elements, and the rest parts are their neighbor contexts.

$\hat{y}_v$'s as the query (with the uncertain element $v$ masked out) to retrieve structurally similar exemplars.

Specifically, we consider the typical setting where there is a retrieval candidates pool $P = \{(x_c, y_c)\}_{c=1}^{|P|}$ which are pairs of input sentences $x_c$ and output graphs $y_c$. To perform structure-aware retrieval, we first prepare a subgraph retrieval pool $P_g = \cup_{c=1}^{|P|} S_d(y_c)$, where $S_d(y_c) = \{y_{c_j}\}_j$ is the set of all depth-$d$ subgraphs of $y_c$. Then, at inference time, given every uncertain subgraph $u_d(v)$, we retrieve $\{y_{c',j'}\}_{c',j'} \subset P_g$ based on their graph similarity with $u_d(v)$, which eventually leads to the set of exemplars $\{(x_{c'}, y_{c'})\}_{c'}$ that will be used for the retrieval-augmented inference.

**Practical Implementation.** In this work, we consider the SMATCH metric (Cai and Knight, 2013) for computing graph similarity. Given a query graph with $N_q$ nodes and $k$ candidate graphs with $N_c$ nodes each, the time complexity of the graph matching algorithm is $O(k * N_q * N_c)$. In practice, the size of $N_q$ is controlled by $d$, and $k$ can be significantly reduced by first pre-filter the candidate pool using a fast heuristic metric (e.g., atom similarity) [3].

### 4.2 Improve Parsing Performance with Uncertainty-aware Iterative Retrieval

Due to its uncertainty-aware nature, SUGAR introduced in Section 4.1 can be applied iteratively to

---

[3]In our experiment, we choose $d = 3$ and retrieve based on 1,000 candidate graph pool, and our exception time is around 5 iterations per seconds.

model prediction, by continuously retrieving new exemplars to address model uncertainty in the previous iteration until model reached a satisfactory level of confidence. This is analogous to the iterative refinement approaches in the recent literature where a model's initial generation can improved by additional self-correction steps (Reid and Neubig, 2022; Schick et al., 2023; Welleck et al., 2023; Jiang et al., 2023).

In the experiment (Section 6.2), we study model performance under different retrieval and refinement strategies, validating that incorporating structure and uncertainty information are both important for improving parsing performance under iterative refinement.

## 5 Experiments

### 5.1 Datasets & Model Settings

**Datasets.** In this paper, we use two complex semantic parsing datasets, including dialogue-oriented semantic parsing and graph-based grammar parsing.

- **SMCalFlow.** SMCalFlow (Andreas et al., 2020) is a large corpus of semantically detailed annotations of task-oriented natural dialogues. The annotation uses dataflow computational graphs, composed of rich set of both general and application specific functions, to represent user requests as rich compositional expressions.

- **Redwoods-Ecommerce (Ecommerce).** The LinGO Redwoods Treebank is a collection of hand-annotated corpora for an English grammar consisting of more then 20 datasets (Oepen et al., 2002). We choose the Ecommerce subset of Redwoods which consists of email messages composed by the online customers. The graph output of this subset is sufficiently complex, and is considered out-of-distribution compared to the standard training split of Redwoods based on Wall Street Journal (Lin et al., 2022b).

**Models & Base Retrievers.** We choose GPT3.5 (text-davinci-003; Ouyang et al., 2022) as our large language models for in-context learning settings. To initialize SUGAR prediction, we consider three choices of base retrievers to be used for initializing SUGAR prediction: (1) Random, (2) CASPER$_{use}$ (Pasupat et al., 2021) that is based on Universal Sentence Encoder (USE) (Cer et al.,

2018), and (3) BM$_{25}$ (Robertson et al., 2009). Recall that these retrievers will only be used to initialize the first round of SUGAR prediction, and not used in subsequent iterations (Algorithm 1).

**Baselines.** Due to the iterative nature of SUGAR, prediction results are comparable to base retrievers mentioned above using the same total number of exemplars. For example, base retriever using 8 exemplars is comparable to SUGAR at iteration 1 (5 base exemplars plus 3 iterative exemplars). However, due to the sequence limitation, it is unfeasible to add large number of exemplars in one prompt, which makes it impossible to get results for base retrievers using more than 10 exemplars. This also highlights the advantage of iterative retrieval as it can get rid of sequence length limitation. Another benefit is that it can get the model's intermediate results towards the final optimal prediction, and more exemplars are added to concentrate on these intermediate results' weak parts.

We also consider two iterative variant of baselines that is based on output similarity: GandR$_{iter}$ (Zemlyanskiy et al., 2022) that retrieves examplars based on BM$_{25}$ with both input sentence and predicted graphs (weight $\alpha$=0.5)[4], and also Oracle that retrieves examplars based on graph similarity with gold subgraphs. Further details about settings of model, candidate pool and prompts can be found in Appendix D, and Appendix E reports additional supplementary studies on much smaller models T5 (Raffel et al., 2020) for out-of-domain and low-resource settings.

### 5.2 Results

The evaluation results are shown in Figure 4. As shown, SUGAR progressively improves the model prediction across iterations, even in the setting where its iterative counterparts becomes counterproductive (e.g, Base=BM$_{2.5}$ on Ecommerce). Specifically, SUGAR significantly outperforms its base retrievers with the same retrieval budget (Base@8), improving the base retriever CASPER$_{use}$ and BM$_{25}$ by 26.61% and 20.36% respectively in absolute percentage on SMCalflow, and 17.58% and 12.37% respectively in absolute percentage on Ecommerce.

In Appendix F, to understand if the benefit of SUGAR is orthogonal to base model choice, and cannot be surpassed by simply upgrading model

---

[4]In ablation analysis Section 6.2, we also consider a variant of GandR that incorporates model uncertainty.

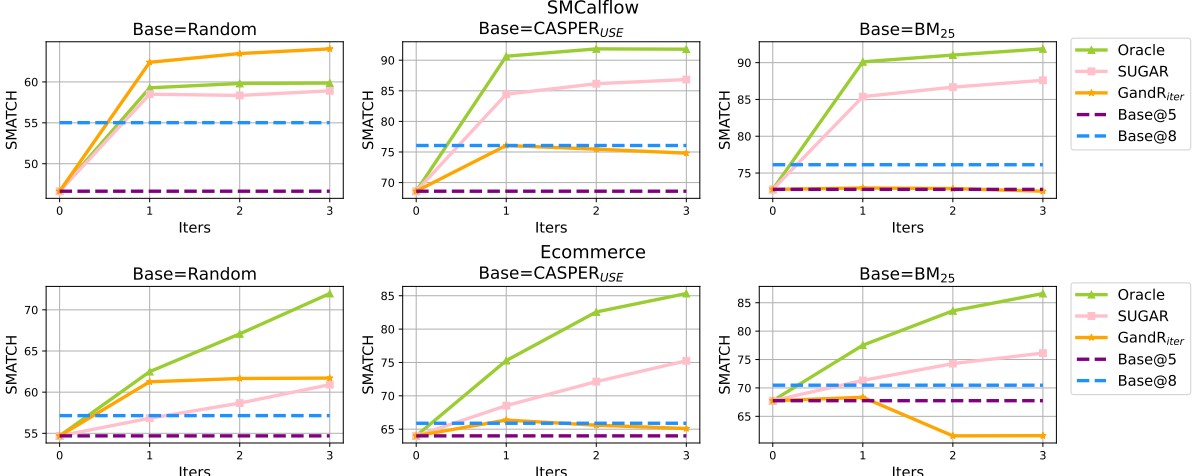

Figure 4: In-context learning results. Base@5 and Base@8 refer to retrieving 5 and 8 exemplars by base retriever. Since iterative retrieval will add 3 additional exemplars per iteration, Base@8 is comparable to results at iter 1.

model architecture, we conduct the following two sets of experiments: (1) a comparison between SUGAR and its baseline variants based on older variants of GPT models (e.g., text-davinci-002) which differ in training method. (2) comparison of baseline methods under GPT-4 v.s. SUGAR under GPT-3.5. The experimental results have shown that SUGAR successfully improves all LLMs in terms of exact match and graph similarities. Furthermore, SUGAR with GPT3.5 also works better than GPT4 using base retriever. This concretely shows our method provides non-trivial improvement beyond what can be achieved by simply upgrading architecture.

Table 2 shows some exemplars (sentence inputs) from different retrievers. We can find that compared to other retrievers, SUGAR can accurately retrieve exemplars that closely align with the uncertain parts of the model prediction, which is significant in addressing the uncertainties that are inherent in the model inference, thereby refining the model's prediction.

## 6 Analysis: the Role of Uncertainty

### 6.1 Uncertainty Quality Matters

We notice that SUGAR does not work very well on the random base retriever, and this might be due to the relatively poor uncertainty quality of the random base retriever in comparison to the other two base retrievers. To validate our hypothesis, we further evaluate the uncertainty quality of the three base retrievers.

A common approach to evaluate model's uncer-

tainty quality is to measure its calibration performance, i.e., whether the model's predictive uncertainty is indicative of the predictive error, e.g., expected calibration error (ECE; Naeini et al., 2015). Based on ECE, Lin et al. (2023) propose Compositional Expected Calibration Error (CECE) to measures the difference in expectation between the model's predictive performance on graph elements and their actual match to the gold graph, which can better reflect model's behavior in predicting graph structures. Table 3 reports the CECE results (based on all graph elements, nodes, and edges).

We can find that the calibration for random base retriever is consistently worse than $BM_{25}$ and $CASPER_{use}$, indicating that the relatively-confident neighbors in uncertain subgraphs for random retriever might fail to capture accurate contexts for iterative retrieval. This also indicates that improving LLMs' calibration can be a fruitful venue for improving retrieval-based data augmentation.

### 6.2 Uncertainty as a Early Stopping Signal

Given that the parsing generation in Figure 1 to edit previous model prediction is a zero-shot process, i.e., the LLM has not been provided with any examples to improve model prediction at each step, thus the model has a tendency to keep making edits to predictions from previous iterations, even if they are already accurate. At this stage, uncertainty becomes a significant signal for stopping the iteration process.

Specifically, SUGAR will terminate the iteration retrieval process and return the last model predic-

| User Input | Retriever | Exemplar User Input |
|---|---|---|
| What is the name of the meeting I have in an hour . | CASPERuse | What time is my last meeting on thursday? |
| | BM$_{25}$ | Please extend the appointment by half of an hour. |
| | GandR | Please extend the appointment by half of an hour. |
| | SUGAR | Fine, then put me down as on vacation starting this second until April 21 |
| get rid of dentist appointment | CASPERuse | Schedule an appointment for the dentist at 2pm tomorrow. |
| | BM$_{25}$ | Dentist appointment is when? |
| | GandR | Dentist appointment is when? |
| | SUGAR | actually can you remove that one |
| what do I have next week that's not a party ? | CASPERuse | what do I have for events tomorrow? |
| | BM$_{25}$ | Do I have a work conference next week? |
| | GandR | Do I have a work conference next week? |
| | SUGAR | Do I have anything else scheduled besides Work and Hustle ? |
| leave early from 3-5 today | CASPERuse | postpone it to the wednesday |
| | BM$_{25}$ | today morning meet |
| | GandR | today morning meet |
| | SUGAR | I need a mechanics appointment today after 4pm |
| Delete my 5 o'clock with simon and the pauls | CASPERuse | Delete my meeting on Saturday |
| | BM$_{25}$ | Take out my 2 o'clock break tomorrow |
| | GandR | Take out my 2 o'clock break tomorrow |
| | SUGAR | Could you please Accept my date with Sam |

Table 2: Exemplars from different retriever on SMCalflow development set. Highlighted text refers to uncertain parts.

| Dataset | Base | CECE$_G$ ↓ | CECE$_N$ ↓ | CECE$_E$ ↓ |
|---|---|---|---|---|
| SMCalFlow | BM$_{25}$ | 0.2617 | 0.2252 | 0.3008 |
| | CASPERuse | 0.2602 | 0.2288 | 0.2943 |
| | Random | 0.3747 | 0.3253 | 0.4297 |
| Ecommerce | BM$_{25}$ | 0.1594 | 0.1119 | 0.2092 |
| | CASPERuse | 0.1677 | 0.1250 | 0.2131 |
| | Random | 0.2000 | 0.1499 | 0.2543 |

Table 3: Compositional ECE for different base retrievers on SMCalflow and Ecommerce under in-context learning settings.

| Dataset | Iter | BM$_{25}$ | | CASPERuse | |
|---|---|---|---|---|---|
| | | GandR | GandR* | GandR | GandR* |
| SMCalFlow | @1 | 71.89 | **72.97** | 76.06 | **77.28** |
| | @2 | 71.83 | **72.89** | 75.46 | **76.77** |
| | @3 | 71.06 | **72.53** | 74.81 | **76.68** |
| Ecommerce | @1 | 68.35 | **69.01** | 66.37 | **66.76** |
| | @2 | 61.56 | **68.41** | 65.60 | **66.34** |
| | @3 | 61.60 | **67.73** | 65.10 | **65.86** |

Table 4: Results of GandR$_{\texttt{iter}}$ baseline in iterative retrieval with and without model uncertainty as the signal for iteration stop. * Means using uncertainty.

tion once a satisfactory level of confidence has been achieved. Additionally, we incorporate model uncertainty as a stopping signal for the GandR$_{\texttt{iter}}$ baseline (Table 4), and we consistently observe an improvement compared to the baseline that doesn't utilize uncertainty as a stopping signal. This further validates the crucial role of uncertainty in the iterative retrieval process.

## 6.3 Uncertainty across Retrieval Iterations

Two additional interesting questions could be (1) what type of graph requires more iterations? and (2) how does uncertainty evolve across different retrieval iterations?

We first explore the correlation between number of iterations and graph complexity and report the results in Table 5 (base retriever is CASPERuse). As we expected, a graph of higher complexity typically demands a greater number of iterations before achieving a satisfactory level.

We then visualize some graphs at different iterations and explore the progression of uncertainty levels (see details in Appendix G). Generally, we notice that an uncertain graph elements will become less uncertain as iterations progress. However, we also observe occasional fluctuations in the uncertainty, which can trigger instability in neighboring contexts. This means that as uncertain elements become certain, the contexts around them may lose some degree of confidence. We reserve further exploration of this phenomenon for future work, which we believe is a promising direction for understanding LLMs' calibration in complex structure generation.

## 7 Conclusions

In this work, we present *Structural-aware and Uncertainty-Guided Adaptive Retrieval* (SUGAR), a new retrieval-augmented parsing framework us-

| Dataset | #Iter | #Node | #Edge | Degree | Depth |
|---------|-------|-------|-------|--------|-------|
| SMCalFlow | 1 | 12.53 | 11.53 | 1.38 | 7.05 |
|           | 2 | 13.16 | 12.16 | 1.43 | 7.39 |
|           | 3 | 16.04 | 15.04 | 1.80 | 9.70 |
| Ecommerce | 1 | 12.68 | 12.53 | 1.89 | 3.32 |
|           | 2 | 14.63 | 14.51 | 1.94 | 3.61 |
|           | 3 | 14.92 | 14.80 | 1.94 | 3.65 |

Table 5: Correlations between number of required iterations verses graph complexity, including number of nodes/edges (#Node/#Edge), average node degrees and graph depth (length of the longest path), where base retriever is CASPER-USE.

ing LLMs for complex graphs. This work deepens the current practice of retrieval-augmented models for complex structures by incorporating information related to the model's uncertainty of graph component prediction and structural similarity of output subgraphs. Experimental results on two complex seq2seq semantic parsing tasks, i.e., SMCalFlow and E-commerce, have demonstrated the practical effectiveness of the proposed approach in the modern setting of graph parsing with pretrained LLMs.

Our future work includes considering more advantage graph similarity metric beyond SMATCH (e.g., incorporates additional similarity metric between the space of node and edge properties), and also larger scale and more fine-grained evaluation on graph parsing benchmarks with distinct properties (e.g., long-tail generalization, graph of specific families) to further elucidate in what setting is the graph similarity most effective. Furthermore, it is also of interest to study the generalization of this approach to a broader class of modern program synthesis problems, e.g., code generation.

## Acknowledgement

Our work is sponsored in part by National Science Foundation Convergence Accelerator under award OIA-2040727 as well as generous gifts from Google, Adobe, and Teradata. Any opinions, findings, and conclusions or recommendations expressed herein are those of the authors and should not be interpreted as necessarily representing the views, either expressed or implied, of the U.S. Government. The U.S. Government is authorized to reproduce and distribute reprints for government purposes not withstanding any copyright annotation hereon. We thank Deepak Ramachandran for helpful discussion.

## Limitations

This work focuses on advanced inference procedures to improve retrieval-augmented LLMs for complex graph parsing problems. The work is limited in two aspects due to the nature of LLM setup: (1) our evaluation has focused on the in-context learning setting where the LLM is not fine-tuned on domain-specific data. Although a standard setting of modern LLMs, it is still of scientific interest to understand the interplay between parameter finetuning and the effectiveness of retrieval-augmentation procedures, which we leave for future work. (2) This work has focused on GPT3.5 which was one of the strongest and openly available LLMs at the time of the writing. As the behavior of LLM can be impacted by its pretraining procedures, it is also of interest to generalize this study to a wider class of LLMs. (3) Finally, the graph similarity metric considered in this work (i.e., SMATCH) has a computational complexity quadratic in graph size. The current work mitigates the issue by restricting its attention to degree-$d$ subgraphs, with the caveat of limiting SUGAR's ability to reason about similarity in the global graph structure. Therefore identifying practical and more computationally efficient structure similarity metrics can further improve the scalability of the SUGAR approach.

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

# A  Graph Matching Algorithm in SMATCH

In general, finding the largest common subgraph is a well-known computationally intractable problem in graph theory. However, for graph parsing problems where graphs have labels and a simple tree-like structure, some efficient heuristics are proposed to approximate the best match by a hill-climbing algorithm (Cai and Knight, 2013). The initial match is modified iteratively to optimize the total number of matches with a predefined number of iterations (default value set to 5). This algorithm is very efficient and effective, it was also used to calculate the SMATCH score in Cai and Knight (2013).

# B  Model Calibration

To explore the model calibration for parsing complex graph structure under in-context learning settings, we plot histgrams to how the correlation between model probability and performance. Specifically, we use base retrievers USE-input, BM25-input and Random, and test them for GPT3.5's in-context learning settings on SMCalflow and Ecommerce respectively. Results are shown in Figure 5. As can be seen from figures, the model is generally calibrated, i.e., high probability generally corresponds to high performance and vice versa. Our

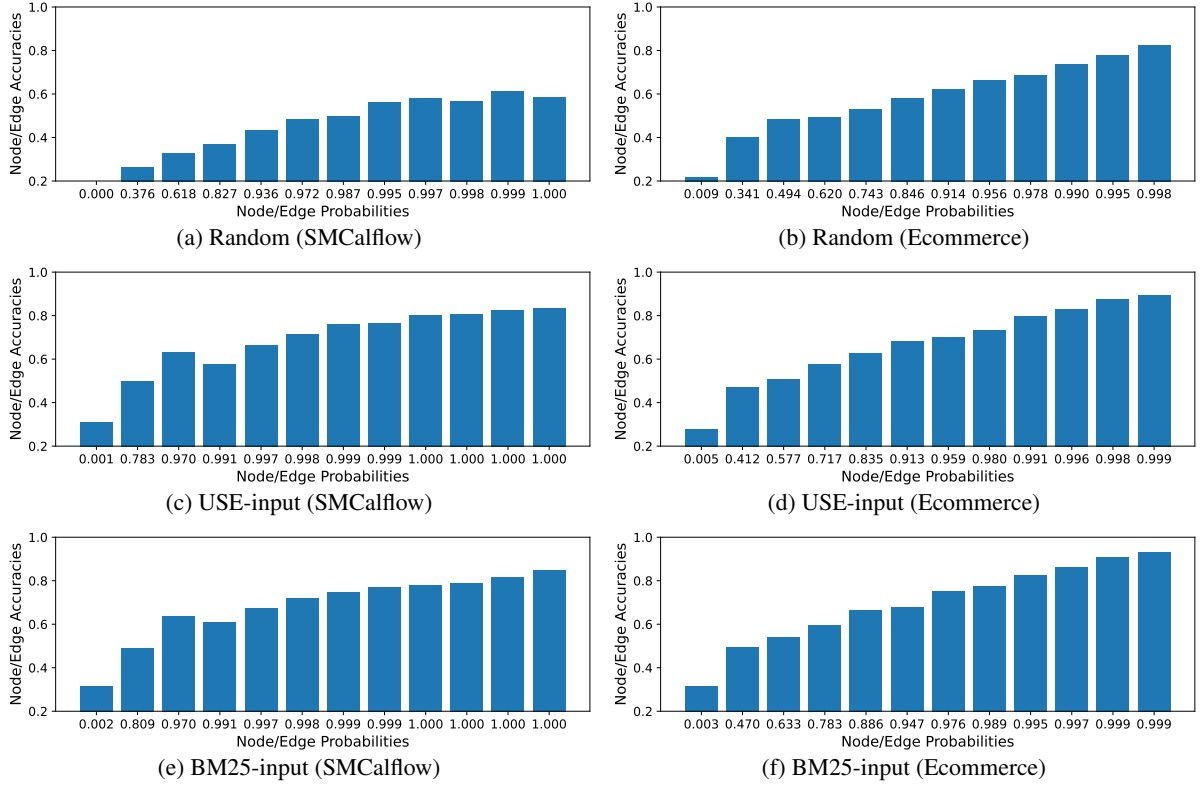

Figure 5: Correlations between model probabilities and performance for node/edge prediction on SMCalFlow and Ecommerce using different retrieval strategies.

study confirms that model uncertainty is effective for detecting prediction errors. This means that we can retrieve structurally similar exemplars targeting on these uncertain substructures, which can help to address the flawed parts in the prediction.

## C  Graph Linearization

We use PENMAN notation for graph linearzation, which is originally called Sentence Plan Notation in the PENMAN project (Kasper, 1989). PENMAN is a serialization format for the directed, rooted graphs used to encode semantic dependencies, mostly notably in the Abstract Meaning Representation (AMR) framework (Banarescu et al., 2013). It looks similar to Lisp's S-Expression in using parentheses to indicate nested structures.

To make PENMAN notation compatible with the seq2seq learning, we adopted a variable-free version of PENMAN which was first proposed in Lin et al. (2022b). Table 6 shows some variable-free PENMAN linearized examples for the two semantic parsing datasets we adopt in our experiments.

## D  Detailed Experiment Settings

**Parameter Settings**  For the uncertain threshold $\epsilon$, considering that at the initial stage, the model's predictions are relatively weak, we set a warm up schedule for $\epsilon$. Specifically, $\epsilon_1 = 0.5, \epsilon_2 = 0.8, \epsilon_3 = 0.9$. We set subgraph max depth $d = 3$. For each model prediction, the number of uncertain subgraphs $k = 3$, and we will retrieve 1 exemplars for each uncertain subgraph.

**Models**  For in-context learning settings, considering the impressive performance achieved by GPT3.5 (Ouyang et al., 2022), we test our methods on text-davinci-003. For fine-tuning settings, we choose T5 (Raffel et al., 2020) as our pretrained model, which is a pre-trained sequence-to-sequence Transformer model that has been widely used in many NLP applications. We use the open-sourced T5X[5], which is a new and improved implementation of T5 codebase in JAX and Flax. Specifically, we use the official pretrained T5-Large (770 million parameters).

---

[5] https://github.com/google-research/t5x

| Datasets | Inputs | Outputs |
|---|---|---|
| SMCalflow | *User: What time on Tuesday is my planning meeting?* | `( start`
`  :ARG1 ( findEvent`
`    :ARG1 ( EventSpec :name "planning"`
`      :start ( Timespec :weekday "tuesday" ) ) ) )` |
| Ecommerce | *You shipped the wrong item.* | `( _ship_v_cause`
`  :ARG1 ( pron`
`    :BV-of ( pronoun_q ) )`
`  :ARG2 ( _item_n_of`
`    :BV-of ( _the_q )`
`    :ARG1-of ( _wrong_a_with ) ) )` |

Table 6: Examples for variable-free PENMAN linearized graph on Ecommerce and SMCalflow (task details can be found in Section 5.1). Here `:carg` means corresponding spans in the sentence.

**Training Data**   For in-context learning, as we access the model through a paid API[6], and there is a limitation for sequence length, we sample a subset of 1,000 test examples from each datasets for test set. For SMCalFlow, we only consider first-turn dialogue in order to reduce sequence length. The candidate pool for SMCalflow is a set of 2,000 examples sampled from the standard training set. The candidate pool for Ecommerce is the standard development set (1.7K examples).

For fine-tuning settings, the training data for SM-CalFlow is a set of 2,000 examples sampled from the standard training set, which also serves as the candidate pool for retrieval, and the test set is the standard development set (15K examples). The training set for Ecommerce is the Redwoods' in-domain dataset on Wall Street Journal (34K examples), the test set and candidate pool are the standard test and development set of Ecommerce (1.1K and 1.7 examples respectively).

**Prompt Design**   There are two prompts adopted in the in-context settings using GPT3.5. The first one contains exemplars from the base retriever to generate the preliminary prediction at the initial stage (prompt 1 shown in Figure 6). The second one contains exemplars from uncertainty-guided retrieval and incorporates the prediction from the previous step (prompt 2 shown in Figure 7).

## E   Fine-tuning Results

We conduct a supplementry evaluation for fine-tuning settings, where we focus on low-resource (SMCalFlow) and out-of-domain (Ecommerce) settings[7]. Specifically, we compare SUGAR with two

---

[6]https://api.openai.com/v1/completions
[7]Since the cost of fine-tuning is higher, we only need one iteration for fine-tuning settings.

---

```
Prompt 1: Exemplars from Base Retriever
Let's translate a sentence input into a structured output.

input: I can't find it anywhere.
output: ( neg :ARG1 ( _can_v_modal :ARG1 ( _find_v_1 :ARG1
        ( pron :BV-of ( pronoun_q ) ) :ARG2 ( pron :BV-of
        ( pronoun_q ) ) :ARG1-of ( loc_nonsp :ARG2 ( place_n
        :BV-of ( _any_q ) ) ) ) ) )
.......
input: I haven't received it yet.
output:
```

Figure 6: Prompt example contains exemplars from base retriever.

baseline retrievers: (1) CASPER using Universal Sentence Encoder (CASPER-USE; Pasupat et al., 2021); (2) GandR considering input and output similarity using BM25 with weight $\alpha = 0.5$ (Zemlyanskiy et al., 2022).

The evaluation results are reported in Table 7. As can be seen, other retrievers that do not consider structural similarity all fail on these two complex parsing tasks, i.e., perform even worse than base model without retrieval, while SUGAR can persistently improve base model on both datasets. Specifically, SUGAR achieves error reduction rate as 6.72% and 9.45% on SMCalflow and Ecommerce respectively, and improves exact match rate by 2.72% and 8.38% respectively.

## F   Results with Other LLMs

The results of SUGAR with `text-davinci-002` and GPT4 are shown in Table 9. We can see that: (1) SUGAR successfully improves all LLMs in terms of exact match and graph similarities. Note that SUGAR Iter 1 is comparable to BM25@8 given the same number of exemplars, and SUGAR Iter 2 and Iter 3 can further improve the results without increasing the general sequence length of the prompt; (2) SUGAR with GPT3.5 also works better than GPT4 using base retriever. This con-

| Name | Retriever | SMCalFlow | | | | Ecommerce | | | |
|---|---|---|---|---|---|---|---|---|---|
| | | EM | Precision | Recall | SMATCH | EM | Precision | Recall | SMATCH |
| Base | - | 60.29 | 83.70 | 86.59 | 85.12 | 51.44 | 92.75 | 94.15 | 93.44 |
| GandR | BM25 | 56.20 | 82.53 | 84.78 | 83.12 | 48.93 | 92.15 | 93.53 | 92.83 |
| CASPER_USE | USE | 55.84 | 82.55 | 84.30 | 83.42 | 49.53 | 92.40 | 93.68 | 93.03 |
| SUGAR | SMATCH | **61.93** | **85.19** | **87.26** | **86.12** | **55.75** | **93.95** | **94.16** | **94.06** |
| Oracle | SMATCH | 64.18 | 86.46 | 88.38 | 87.39 | 56.73 | 94.08 | 94.48 | 94.28 |

Table 7: Fine-tuning results. EM means exact match rate. Precision and recall are based on graph element triples.

---

**Prompt 2: Exemplars from Iterative Retrieval**

```
Let's fix a model output ( output* ) based on some examples
if given.

input: have you recieved the Siemens S40 yet?
output: ( _receive_v_1 :ARG1 ( pron :BV-of ( pronoun_q ) )
        :ARG2 ( named :BV-of ( _the_q ) :ARG1-of ( compound
        :ARG2 ( named :BV-of ( proper_q ) :carg " Siemens " ) )
        :carg " S40 " ) :ARG1-of ( _yet_a_1 ) ) )
.......

input: I haven't received it yet.
output*: ( neg :ARG1 ( _receive_v_1 :ARG1 ( pron :BV-of
        ( pronoun_q ) ) :ARG2 ( pron :BV-of ( pronoun_q ) )
        :ARG1-of ( _yet_r ) ) )
output:
```

Figure 7: Prompt example contains exemplars from iterative retriever.

node `pron` from the first to second iteration in the second example (the probability decreases from 0.9381 to 0.8746). We reserve this observation for further exploration in future work, which we believe is a promising direction for understanding LLMs' calibration in complex structure generation.

cretely shows our method provides non-trivial improvement beyond what can be achieved by simply upgrading architecture. We will experiment with SUGAR+GPT4 in future work (cannot do it right now due to API limitations).

| Model | Retriever | EM | SMATCH |
|---|---|---|---|
| GPT-4 | BM25@8 | 33.20 | 78.21 |
| text-davinci-002 | BM25@8 | 27.00 | 73.28 |
| text-davinci-002 | SUGAR Iter 1 | **43.30** | **83.06** |
| text-davinci-002 | SUGAR Iter 2 | **52.70** | **83.76** |
| text-davinci-002 | SUGAR Iter 3 | **53.50** | **84.42** |

Table 9: SUGAR with `text-davinci-002` and GPT4 on SMCalflow task.

## G   Sample Graph Prediction Visualizations

Some sample graph prediction visualizations on Ecommerce dataset using CASPER-USE are shown in Table 8. We can observe that as iteration goes, the confidence scores of uncertain graph elements generally increase and the number of uncertain graph elements generally decreases.

However, we also observe occasional fluctuations in the uncertainty, which can trigger instability in neighboring contexts. For example, the

| Sentence | I have received two shipments from you for one order. |
| --- | --- |
| Iter1 | 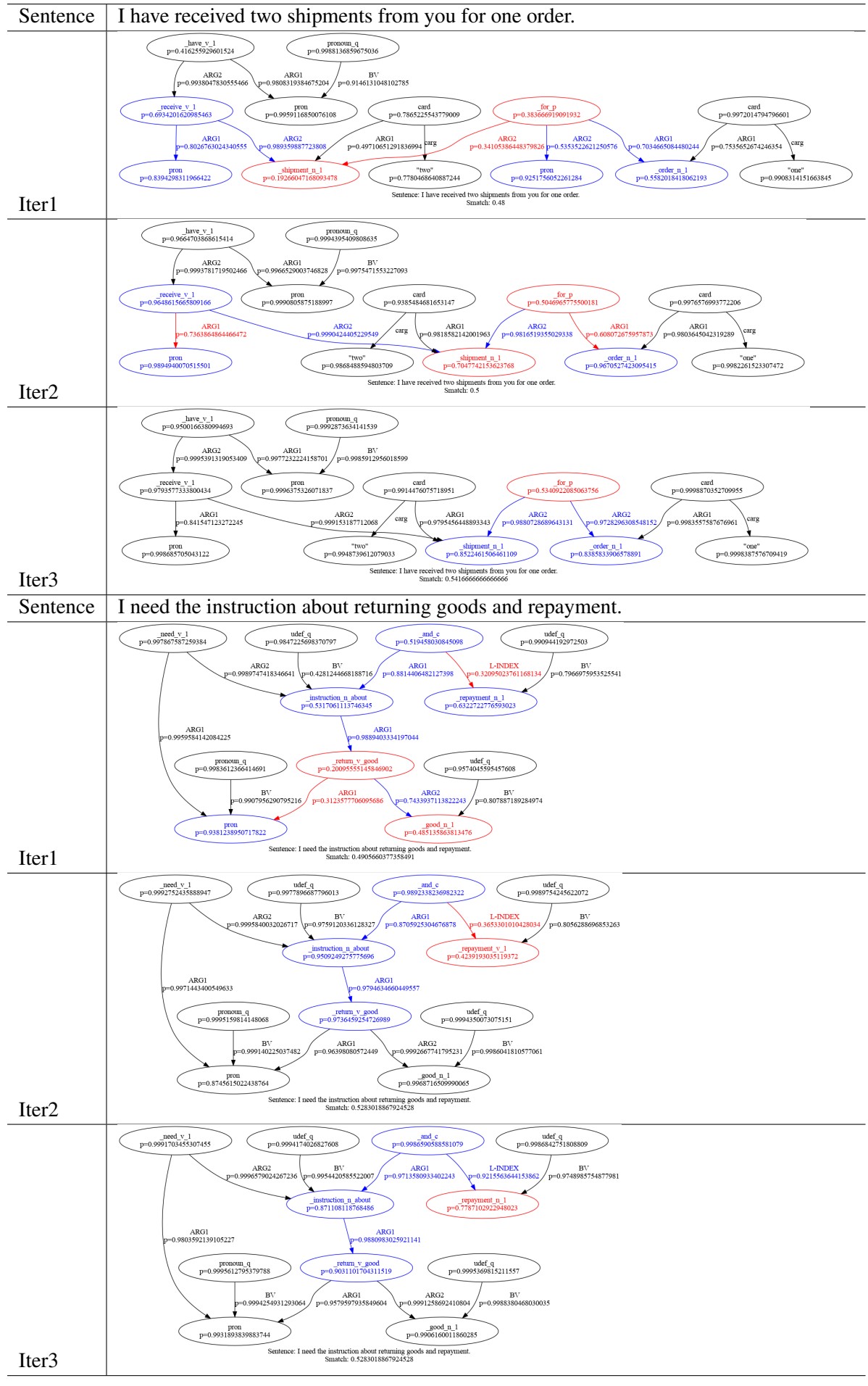 |
| Iter2 | |
| Iter3 | |
| Sentence | I need the instruction about returning goods and repayment. |
| Iter1 | |
| Iter2 | |
| Iter3 | |

| Sentence | You sent me the wrong camcorder! |
|----------|----------------------------------|
| Iter1 | 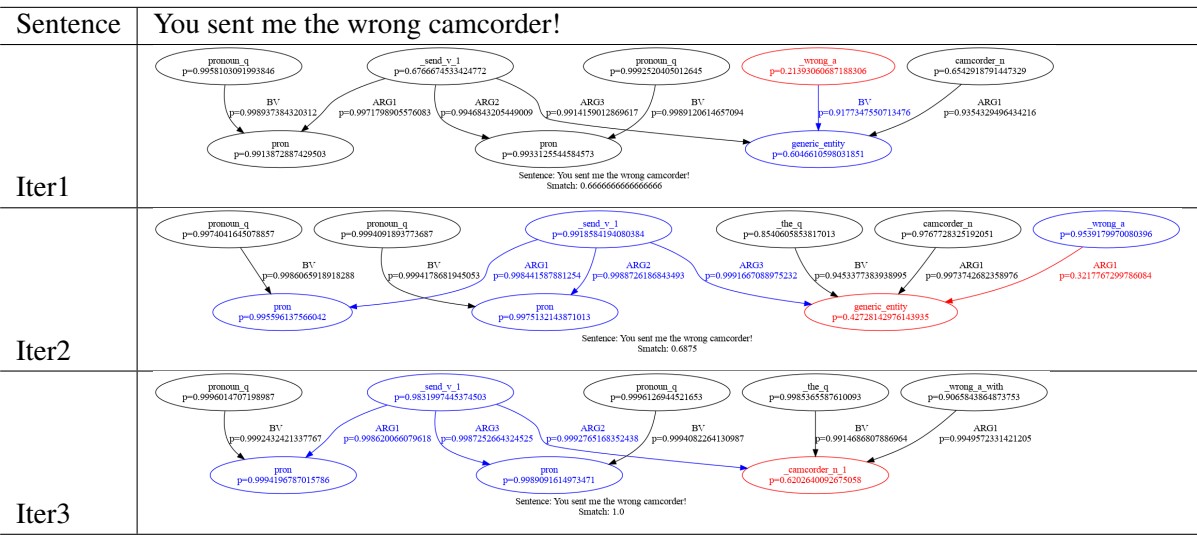 |
| Iter2 | |
| Iter3 | |

Table 8: Sample graph prediction visualizations for iterative progression. Red refers to uncertain graph elements and blue refers to confident neighbor contexts.