# OpenReview forum: "Retrieval-Augmented Parsing for Complex Graphs by Exploiting Structure and Uncertainty"
_EMNLP/2023/Conference — EMNLP 2023 Findings_

### Official Review · Reviewer_Gxdf · 2023-07-24

**Soundness:** 3

**Excitement:**

3: Ambivalent: It has merits (e.g., it reports state-of-the-art results, the idea is nice), but there are key weaknesses (e.g., it describes incremental work), and it can significantly benefit from another round of revision. However, I won't object to accepting it if my co-reviewers champion it.

**Paper Topic And Main Contributions:**

This paper is about retrieval augmented semantic parsing using LLM. It uses iterative retrieval, where each iteration conducts uncertainty estimation and uses the neighbors of the uncertain nodes as the input for the additional retrieval.
For the uncertainty estimation, Graph Autoregressive Process (GAP) is used instead of just the confidence of the graph element.
SMATCH is used as the semantic retrieval evaluation metric.

They showed the proposed iterative retrieval approach's effectiveness both quantitatively and qualitatively.

Some issues are:
1. It is not shown whether GAP is significantly better than the confidence of the graph element for the confidence estimation. An ablation study should be included.
2. Using uncertainty for the retrieval decision or iterative retrieval are not novel ideas. FLARE, which was also cited, is an example.
3. As LLM is used, generative multi-hop retrieval methods could be more effective than the proposed method. It is uncertain whether this would be the most effective approach on top of LLMs.

**Reasons To Accept:**

They showed the proposed iterative retrieval approach's effectiveness both quantitatively and qualitatively.
The approach is suitable especially for the semantic parsing, which is with different properties compared to general iterative retrievals. Using GAP for the uncertainty estimation and SMATH as the metric look proper decisions.

**Reasons To Reject:**

1. It is not shown whether GAP is significantly better than the confidence of the graph element for the confidence estimation. An ablation study should be included.
2. Using uncertainty for the retrieval decision or iterative retrieval are not novel ideas. FLARE, which was also cited, is an example.
3. As LLM is used, generative multi-hop retrieval methods could be more effective than the proposed method. It is uncertain whether this would be the most effective approach on top of LLMs.

**Reproducibility:**

4: Could mostly reproduce the results, but there may be some variation because of sample variance or minor variations in their interpretation of the protocol or method.

**Reviewer Confidence:**

4: Quite sure. I tried to check the important points carefully. It's unlikely, though conceivable, that I missed something that should affect my ratings.

---

> ### Author Rebuttal · Authors · 2023-08-28
>
> Dear Reviewer Gxdf,
>
> We really appreciate your comments. We hope our following point-to-point response can address your concerns.
> 1. *It is not shown whether GAP is significantly better than the confidence of the graph element for the confidence estimation. An ablation study should be included.*
> - GAP is the first probabilistic framework that can map sequence-level probabilities to graph-level probabilities for quantifying a seq2seq model’s structural uncertainty on graph elements. Given that we need to consider structural uncertainty in our problem settings, GAP is the only framework we can adopt for uncertainty quantification at the graph level.
> - To address your concern, we further conduct an ablation study using only token-level uncertainty and retrieve uncertain sub-sequence of the output using token-level similarity BM25, following similar settings as SUGAR. This setting is the same as FLARE without considering any structural information (so we termed it FLARE here). The results on SMCalflow at the first iteration are shown as follows:
> | Model  | EM        | Precision | Recall    | SMATCH    |
> |--------|-----------|-----------|-----------|-----------|
> | BM25@8 | 31.80     | 71.28     | 81.69     | 76.13     |
> | FLARE  | 32.70     | 74.54     | 82.18     | 78.17     |
> | SUGAR  | **49.80** | **80.48** | **90.93** | **85.38** |
> - As shown in the table, FLARE works slightly better than baseline model BM25@8 using the same number of retrieved exemplars but works much worse than SUGAR. This is because those uncertain sub-sequences do not necessarily correspond to a meaningful sub-graph structure, and will be difficult for a retriever to find an informative candidate, e.g., for a graph (tree) prediction as:
> ```
>     A
>    / \
>   B   C
>  / \
> D   E
> ```
> - which is linearized as A(B(D,**E**),C). If the uncertain graph element is **E**, for SUGAR, the uncertain subgraph is B(D,**E**), while for FLARE, the uncertain sub-sequence is (D,**E**),C, which are all leave nodes that have no direct dependencies to form an informative context for retrieval.
>
> 2. *Using uncertainty for the retrieval decision or iterative retrieval are not novel ideas. FLARE, which was also cited, is an example.*
> - The paper focuses on the joint exploitation of two sources of information: (1) structural similarity; (2) structural uncertainty.
> - FLARE is a concurrent work to ours, and is a reduced variant of our work. It only considers token-level similarity (BM25) and token-level uncertainty (FLARE = SUGAR without structural information), which is fine for question-answering tasks (where the output is naturally a sequence). However, for parsing complex structured outputs, simply considering token-level information is behind satisfaction (as shown in Table 1). Furthermore, in the above ablation study, we have shown that FLARE could fail to retrieve informative graphs compared to SUGAR. As a result, FLARE can be less effective than SUGAR in exploiting structured information for graph problem domains.
>
> 3. *As LLM is used, generative multi-hop retrieval methods could be more effective than the proposed method. It is uncertain whether this would be the most effective approach on top of LLMs.*
> - Thanks for this question. Indeed, [generative multi-hop retrieval (GMR)](https://aclanthology.org/2022.emnlp-main.92/) bears some similarities to our work in that it performs iterative retrieval, and the retrieval function itself relies on LLM. While GMR can be a powerful method for non-structured problems (e.g., QA) with LLM, it is less suitable for the current setting we focus on (i.e., unsupervised retrieval in graph problem domain). The main reasons are two: (1) to the knowledge of the authors, the retrieval model $p_{LLM}$ (candidate graph | query graph) that GMR relies on needs to be trained; (2) similar to FLARE, GMR is also not designed to exploit structural information, which can be less effective for graph parsing problems.
> - Thanks again for mentioning this relevant work, we will include generative multi-hop retrieval as a related work in Section 2 and include the above discussion.

---

### Official Review · Reviewer_GY72 · 2023-08-02

**Soundness:** 3

**Excitement:**

3: Ambivalent: It has merits (e.g., it reports state-of-the-art results, the idea is nice), but there are key weaknesses (e.g., it describes incremental work), and it can significantly benefit from another round of revision. However, I won't object to accepting it if my co-reviewers champion it.

**Paper Topic And Main Contributions:**

topic: retrieval-augmented LLM
main contribution:
1) Conducting research work demonstrates that relying solely on input text for knowledge retrieval is not reasonable in the semantic parsing task.
2) Introduce the SUGAR method as a solution to overcome the previous limitation.

**Reasons To Accept:**

Conducting research work demonstrates that relying solely on input text for knowledge retrieval is not reasonable in the semantic parsing task. I think this point is important


**Reasons To Reject:**

you should give a detailed description of Figure 3
line 333-line 343, what is the difference between yc and v, and how to do the retrieve, using uncertain subgraph to....?
please give an intuitive description of the retrieval process. If the yc is the generated graph, using an uncertain subgraph to retrieve the generated graph, I am not sure if this method is right or wrong

**Reproducibility:**

3: Could reproduce the results with some difficulty. The settings of parameters are underspecified or subjectively determined; the training/evaluation data are not widely available.

**Reviewer Confidence:**

3: Pretty sure, but there's a chance I missed something. Although I have a good feel for this area in general, I did not carefully check the paper's details, e.g., the math, experimental design, or novelty.

---

> ### Author Rebuttal · Authors · 2023-08-28
>
> Dear reviewer GY72,
>
> We hope our response can address your concerns.
>
> **Regarding $y_c$ and $v$:**
> - $y_c$: candidate graphs in the candidate pool, which is used for retrieving the exemplars. $y_c$ is a gold graph, not a generated graph.
> - $v$: the uncertain graph element from the preliminary prediction, e.g., the red square in Figure 3.
>
> We will include a more detailed explanation in the caption of Figure 3 to make it more understandable.
>
> **Intuitive description of the retrieval process:**
>
> The overview of the retrieval process is shown in Figure 1, which includes the following steps:
> - **Graph uncertainty quantification for preliminary prediction (Section 4.1.1):** We will quantify the uncertainty of graph element $v$ based on the prediction using a base retriever (a retriever that only considers input sentence similarity). This step can help us to locate the uncertain parts of the generated graph.
> - **Uncertain subgraph construction (Section 4.1.2):** An uncertain subgraph (dotted square in Figure 3), which includes uncertain element $v$ (red square in Figure 3) and relatively confident neighbors (blue square in Figure 3), will serve as a query in the retrieval process. The intuition is that, given the model is relatively calibrated, the uncertain parts are likely to be incorrect, and the relatively confident neighbors are likely to be correct, so **we can utilize the uncertain subgraph to retrieve a similar gold subgraph that can help the model to address the uncertain parts**. Considering uncertain subgraphs instead of the entire graph can also reduce the computational complexity of structural similarity measurement.
> - **Structural-aware Retrieval (Section 4.1.3)**: To identify informative graph exemplars that best address the model uncertainty in predicting graph elements, **we leverage the uncertain subgraph mentioned above and consider a retrieval policy using uncertain subgraph as a query and retrieve structurally similar exemplars in candidate pool consisting of candidate graphs $y_c$**. The motivation for using structural similarity has been illustrated in Section 3.1, where we have shown that for complex parsing problems, exemplars that are similar in structure can help more for in-context learning compared to exemplars similar in input sentences or similar in output sequence (e.g., using BM25).
>
> The above process can be operated in an iterative manner to progressively improve the generation performance.

---

### Official Review · Reviewer_yeis · 2023-08-16

**Soundness:** 3

**Excitement:**

4: Strong: This paper deepens the understanding of some phenomenon or lowers the barriers to an existing research direction.

**Paper Topic And Main Contributions:**

Authors of this paper propose a novel approach called Structure-aware and Uncertainty-Guided Adaptive Retrieval (SUGAR) that leverages both structural similarity and model uncertainty to improve retrieval-augmented parsing for complex graph problems. It works towards refining the model’s prediction through exemplars retrieval that closely align with the uncertain parts of the model prediction. They apply the solution on two real-world complex graph parsing benchmarks showing that SUGAR outperforms their classical counterparts that do not leverage uncertainty or structure.

**Reasons To Accept:**

1.	The research idea of the paper is interesting (leverages both structural similarity and model uncertainty).
2.	Paper is crisp, well-structured and detailed.
3.

**Reasons To Reject:**

1.	Evaluation can be more rigorous across various tasks.
2.	Computational complexity needs to be reduced across the abord esp. given the context of graph parsing problems.
3.	The evaluation of the proposed approach has focused on the in-context learning setting where the LLM is not fine-tuned on domain-specific data.
4.	The choice of LLM itself is can have implications when upgraded to more latest architectures (beyond GPT 3.5). (behavior of LLM can be impacted by its pre-training procedures and generalization with wider class of LLMs can be an open question)
5.	Clarity around assumptions on distribution of the data being representative or not-representative of the retrieved exemplars may help.
6.	Details around computational complexity & cost of the retrieval process or the overall complexity of the approach can help.


**Reproducibility:**

3: Could reproduce the results with some difficulty. The settings of parameters are underspecified or subjectively determined; the training/evaluation data are not widely available.

**Reviewer Confidence:**

3: Pretty sure, but there's a chance I missed something. Although I have a good feel for this area in general, I did not carefully check the paper's details, e.g., the math, experimental design, or novelty.

---

> ### Author Rebuttal · Authors · 2023-08-28
>
> Dear Reviewer yeis,
>
> We really appreciate your comments. We hope our following point-to-point response can address your concerns.
>
> 1. *Evaluation can be more rigorous across various tasks.*
> - We evaluate different aspects of model behavior under a variety of controlled settings across two tasks of varying complexity in terms of linguistic diversity and complexity of the output, including
>     - **Retrieval performance:** As measured by similarity to gold graph (StoO in Table 1) and case study of retrieved examples (Table 2).
>     - **Model calibration:** As measured by correlations between model probabilities and performance (Figure 2), and compositional ECE (Table 3).
>     - **Final prediction performance**: as measured by sequence exact match and graph similarity (EM and SMATCH in Table 1 and Figure 4)
> - Understandably, there are always spaces for further improvement of our evaluations, and we will value any suggestions on the specific aspects of the evaluation for improvement.
>
> 2. *Computational complexity needs to be reduced across the abord esp. given the context of graph parsing problems.*
> - Yes, computational complexity is especially important for graph parsing problems. As can be seen in Section 4.1.3 (line 345-354), we have proposed a practical implementation with (1) retrieval based on uncertain subgraphs and (2) pre-filtering for the candidate pool to reduce complexity. We have included a detailed discussion of the theoretical complexity in this section.
> - Also please note that the framework we proposed is general and independent of specific algorithmic implementations. Although we selected relatively classic algorithms (SMATCH) in this work to ensure consistency with the literature, the proposed framework can always benefit from more advanced algorithms in the future.
>
> 3. *The evaluation of the proposed approach has focused on the in-context learning setting where the LLM is not fine-tuned on domain-specific data.*
> - Thanks for this question. In fact, we have also investigated the performance of SUGAR in the fine-tuning settings (see Appendix D&E), where we find that SUGAR can also improve the performance of the base model and outperform other retrievers (GandR and CASPER_USE) on both SMCalflow and Ecommerce datasets.
>
> 4. *The choice of LLM itself is can have implications when upgraded to more latest architectures (beyond GPT 3.5). (behavior of LLM can be impacted by its pre-training procedures and generalization with wider class of LLMs can be an open question)*
> - We adopt two LLMs in the paper, one is GPT3.5 (text-davinci-003) for in-context learning setting (Section 5.1), and the other is T5 for fine-tuning setting (Appendix D&E), and both can achieve performance improvements. Unfortunately, OpenAI API does not support token-level probabilities for models beyond GPT3.5, which makes it not possible to experiment with uncertainty-based methods with GPT-4.
> - Alternatively, to answer the reviewer's question regarding the impact of the base pre-trained model on method performance, we conducted the following two sets of experiments: (1) a comparison between SUGAR and its baseline variants based on older variants of GPT models (e.g., text-davinci-002) which differ in training method. (2) comparison of baseline methods under GPT-4 v.s. SUGAR under GPT-3.5, to understand if the benefit of SUGAR is orthogonal to base model choice, and cannot be surpassed by simply upgrading model model architecture.
> - All the results are summarized as follows (using base retriever BM25 for SMCaflow task):
>
> | Model            | Retriever    | EM        | Precision | Recall    | SMATCH    |
> |------------------|--------------|-----------|-----------|-----------|-----------|
> | GPT-4            | BM25@8       | 33.20     | 74.57     | 82.22     | 78.21     |
> |                  |              |           |           |           |           |
> | text-davinci-002 | BM25@8       | 27.00     | 68.74     | 78.46     | 73.28     |
> | text-davinci-002 | SUGAR Iter 1 | **43.30** | **78.94** | **87.63** | **83.06** |
> | text-davinci-002 | SUGAR Iter 2 | **52.70** | **79.58** | **88.40** | **83.76** |
> | text-davinci-002 | SUGAR Iter 3 | **53.50** | **79.95** | **89.41** | **84.42** |
> |                  |              |           |           |           |           |
> | text-davinci-003 | BM25@8       | 31.80     | 71.28     | 81.69     | 76.13     |
> | text-davinci-003 | SUGAR Iter 1 | **49.80** | **80.48** | **90.93** | **85.38** |
> | text-davinci-003 | SUGAR Iter 2 | **56.00** | **81.91** | **92.00** | **86.67** |
> | text-davinci-003 | SUGAR Iter 3 | **59.10** | **83.37** | **92.28** | **87.60** |
> |                  |              |           |           |           |           |
> | T5               | GandR@3      | 56.20     | 82.53     | 84.78     | 83.12     |
> | T5               | CASPER_USE@3 | 55.84     | 82.55     | 84.30     | 83.42     |
> | T5               | SUGAR@3      | **61.93** | **85.19** | **87.26** | **86.12** |
>
> - As can be seen from the table:
>     - SUGAR successfully improves all LLMs in terms of exact match and graph similarities. Note that SUGAR Iter 1 is comparable to BM25@8 given the same number of exemplars, and SUGAR Iter 2 and Iter 3 can further improve the results without increasing the general sequence length of the prompt.
>     - SUGAR with GPT3.5 also works better than GPT4 using base retriever. This concretely shows our method provides non-trivial improvement beyond what can be achieved by simply upgrading architecture. We will experiment with SUGAR+GPT4 in future work (cannot do it right now due to API limitations).
>     - For fine-tuning since the training process is expensive compared to in-context learning, we only do one round of retrieval, and it can consistently outperform other retrievers.
>
> 5. *Clarity around assumptions on distribution of the data being representative or not-representative of the retrieved exemplars may help.*
> - Regarding the data distribution of the retrieval candidate pool and the test data, we make the standard assumptions that the distribution of the retrieval candidate pool is representative of the distribution of the test data (within the same domain), which is adapted in many retrieval related work ([Pasupat et al.](https://doi.org/10.18653/v1/2021.emnlp-main.607), 2021; [Gupta et al.](https://doi.org/10.18653/v1/2022.nlp4convai-1.15), 2022; [Zemlyanskiy et al.](https://aclanthology.org/2022.coling-1.438), 2022).
>
> 6. *Details around computational complexity & cost of the retrieval process or the overall complexity of the approach can help.*
> - Please refer to discussion in point 2.

---

### Meta-Review · Area_Chair_6hTM · 2023-09-18

**Recommendation:** 4

**Metareview:**

This paper presents a hybrid method to improve graph-based parsing in realistic contexts. A number of experimental choices could be explained and motivated better. However, the proposal is considered valid and the task relevant.

---

### Decision · Program_Chairs · 2023-10-07

**Decision:**

Accept-Findings

**Comment:**

This paper presents a hybrid method to improve graph-based parsing in realistic contexts. A number of experimental choices could be explained and motivated better. However, the proposal is considered valid and the task relevant.